# Motor Coordination and Grip Strength of the Dominant and Non-Dominant Affected Upper Limb Depending on the Body Position—An Observational Study of Patients after Ischemic Stroke

**DOI:** 10.3390/brainsci12020164

**Published:** 2022-01-26

**Authors:** Anna Olczak, Aleksandra Truszczyńska-Baszak

**Affiliations:** 1Rehabilitation Clinic, Military Institute of Medicine, 04-141 Warsaw, Poland; 2Faculty of Rehabilitation, Józef Piłsudski University of Physical Education, 00-968 Warsaw, Poland; aleksandra.truszczynska@awf.edu.pl

**Keywords:** dominant hand, non-dominant hand, dominant wrist, non-dominant wrist, motor coordination, grip force, stabilization

## Abstract

Stroke is one of the leading causes of human disability globally. Motor function deficits resulting from a stroke affect the entire body, but relatively often it is the upper limbs that remain ineffective, which is very limiting in everyday life activities. The finding in neurorehabilitation that trunk control contributes to upper limb function is relatively common but has not been confirmed in clinical trials. This observational prospective study aims to analyze the effect of the position of the trunk and the affected upper limb on the coordination and grip strength of the affected dominant and non-dominant hand and wrist in people after ischemic stroke. The research was carried out at the Department of Neurological Rehabilitation, on a group of 60 patients with acute ischemic stroke. A Hand Tutor device and a hand dynamometer were used for the main measurements of the motor coordination parameters (maximum range of motion, frequency of movement) and the grip strength of the dominant and non-dominant upper limb. The patients were examined in two positions: sitting without back support and lying on the back with stabilization of the upper limb. Higher and relevant results were observed in the non-dominant hand, in the supine position in terms of motor coordination parameters of the fingers (*p* = 0.019; *p* = 0.011) and wrist (*p* = 0.033), and grip strength (*p* = 0.017). Conclusions: The laying position and stabilization of the affected upper limb in the acute phase following ischemic stroke is more beneficial for the coordination of movements and grip strength of the non-dominant hand.

## 1. Introduction 

Lateralization, or sidedness or functional asymmetry of the right and left sides of the human body, results from differences in the structure and functions of the cerebral hemispheres, and is a consequence of the domination of one of the hemispheres [1]. It is expressed, for example, by greater mobility of the right limbs than the left ones, in addition to the registration by the brain of a greater number of sensory stimuli coming from one side of the body [2,3]. Lateralization is a progressive process; it develops gradually with age and overall motor development, but motor dominance remains constant throughout adult life [4].

Human mobility is based on different functions of the upper and lower limbs, and the left and right limbs (functional asymmetry) [2,5]. The specialization and coordination of hand movements is based on the different movements of the hands. The leading hand (right-handed, left-handed) performs the main activity, while the subordinate hand supports it. Coordination of the leading and subordinate hands allows a high degree of dexterity to be achieved, so a high level of perception and economy of movements can be achieved when one of the limbs dominates and the other supports it and cooperates with it [6]. Moreover, when the dominant hand is on the same side of the body as the dominant eye, the so-called hand–eye system, which is the basis of eye–hand coordination, enables and greatly facilitates the performance of graphic and manipulative activities [7,8,9]. 

The literature on the subject describes many different types of coordination related to the movement abilities of the human body (neuromuscular, inter-muscular and intramuscular coordination, eye–hand coordination, hand coordination, etc.). There are also many definitions for the coordination of movement as measured by the movement of body parts. One of these states that motor coordination is a combination of body movements caused by kinematic parameters (e.g., spatial direction) and kinetic (force) parameters, which result in the intended actions [10,11]. Coordinated movement is characterized by appropriate speed, distance, direction, synchronization, and muscle tension [12].

It is also known that the stabilization of the human body is the basis for maintaining balance and enables the performance of selective, coordinated movements of the body [13,14,15]. The ability to rhythmically synchronize moving limbs and limb segments is one of the most basic abilities of vertebrate and invertebrate movement systems [16]. As noted by Kelso (1994), these abilities are the main expression of how movements are organized in time and space, and allow the body to meet the competitive challenges of stability and flexibility [17]. Furthermore, it seems that a stable trunk is the most important element of the body posture control mechanism [18,19,20]. It has been shown that the stability of the spine or the trunk influences the motor coordination of hip and shoulder flexion on the side without paresis, in addition to flexion and extension of the trunk in both patients with chronic stroke and healthy subjects [21]. Trunk control is believed to be important for the mobility of the upper limb, although this assumption has not been thoroughly demonstrated in clinical trials. Researchers long ago noticed a relationship between the position of the trunk and the ability to move the upper limb or the position of the upper limb for the movement of their distal part. For example, Souque’s phenomenon, discovered in 1916, described in Brunstrum’s book (1970), occurs when the elevation of the affected arm causes the extension of the paralyzed hand fingers [22]. In turn, control of the elbow joint, as stated by Ellis et al., depends on the degree of shoulder abduction, and the force of ab-duction is directly related to the moment of flexion in the elbow and, consequently, affects the range of motion in patients after stroke [23]. Wee et al. examined the significance of trunk support in the lumbar region in a sitting position for the functional abilities of the upper limb and obtained confirmation of their hypothesis [24].

In our work, research was conducted on a group of patients after ischemic stroke. Ischemic stroke is the most common type of stroke. Obstruction of a blood vessel supplying the brain results either from a clot that has traveled in the blood from another part of the body (an embolism) or from a clot that has formed locally (a thrombus) [25].

In patients after a stroke, the tension and strength of the superficial and deep muscles stabilizing the trunk are most often reduced. This leads to asymmetries and inappropriate movement patterns. In the acute phase of stroke, most patients have impaired efficiency of the upper limb [26], and in the chronic phase more than half of patents still have a deficit in upper limb mobility [27,28,29]. Stroke patients in the acute phase are often unable to sit properly, even if supported. In addition, it is not possible to wait for the return of the trunk function before commencing work with the affected upper limb in high positions that are more similar to those in everyday activities.

As demonstrated by Gauthier et al. using Constraint-Induced Movement therapy, rehabilitation of the affected upper limb is very important. This involves a structural reconstruction of the human brain, which results in functional improvement, even in adult, chronic stroke patients [30]. Moreover, the latest studies show that, in order to improve the functioning of the hand, one should strive to maintain and restore equal results in terms of the grasping and squeezing force of the dominant and non-dominant hands [31,32]. Researchers suggest that upper limb rehabilitation exercises after stroke should involve both dominant and non-dominant upper limbs [33].

Therefore, it seems important to investigate whether the lying position of the trunk and passive stabilization of the affected upper limb in patients in the acute phase of a stroke may yield better results in terms of motor coordination and hand-grip strength.

The aim of this study was to analyze the influence of the position of the trunk and the affected upper limb on the motor coordination and grip strength of the dominant and non-dominant hands and wrists in patients after ischemic stroke in the acute phase of the disease.

## 2. Methods

### 2.1. Study Design

This was an observational prospective study. The observation concerned the influence of the measurement position on the parameters of motor coordination and the grip strength of the dominant and non-dominant upper limb in patients after stroke. The assessment of the effects of the intervention was undertaken in two different starting positions: sitting without support, and lying with the stabilization of the examined, affected upper limb (independent variable). The effect of the intervention, improvement in motor coordination parameters, and hand-grip strength (dependent variables) were assessed with a Hand Tutor device and an electronic hand dynamometer.

### 2.2. Ethics

This study was carried out in accordance with the recommendations of the Ethical Committee of the Military Medical Institute (MMI) in Warsaw, Poland, which approved the protocol (approval number 4/MMI/2020). Prior to inclusion, all subjects were informed about the purpose of the study. Written informed consent was obtained from all subjects in accordance with the tenets of the Declaration of Helsinki.

### 2.3. Subjects

In total, 100 people were examined before inclusion. Forty people were excluded; 35 patients were excluded because of their functional condition, and 5 declined to participate. Finally, 60 study participants were prospectively placed into two groups: 30 post-stroke with a dominant upper limb and 30 with a non-dominant upper limb (Figure 1).

The study group consisted of 60 post-stroke (thromboembolic, ischemic) patients (30 males and 30 females) recruited from among patients of the Teaching Department of Rehabilitation of the Military Medical Institute (WIM) (aged 22–90 years; mean, 65.3 ± 14.43 years). They were 5–8 weeks post-stroke. They had a stable trunk (a Trunk Control Test score of 74–100 points) and were in a functional state that allowed movements of the upper extremity (FMA-UE 43–49 motor function points), and muscle tension measured by the Modified Ashworth Scale was MAS ≤ 1/1+ [34,35,36]. The clinical evaluation of patients after a stroke was performed by the physician admitting the patient to the clinic on the day of admission. The basic epidemiological data of the study population and the clinical control group are presented in Table 1, and biometric data of the study population and the clinical control group are presented in Table 2.

Criteria for stroke group inclusion were: (1) participants with cerebral ischemic stroke; (2) participants with hemiparesis after 5 to 7 week after stroke; (3) participants with a stable trunk (TCT 70–100 points); (4) participants with functional state allowing movements of the upper extremity (FMA-UE 40–66 motor function points); (5) muscle tension (MAS ≤ 0 −1+); (6) no severe deficits in communication, memory, or understanding that could impede proper measurement performance; (7) at least 20 years of age; maximum 90 years of age.

Exclusion criteria were: (1) stroke up to five weeks after the episode; (2) other than cerebral ischemic stroke; (3) lack of trunk stability; (4) no wrist and hand movement; (5) muscle tension >2 MAS; (6) high or very low blood pressure; (7) dizziness or malaise of the respondents.

### 2.4. Measurements

The research was carried out according to the protocol no 7/KRN/2019.

A Hand Tutor device and an EH 101 electronic hand dynamometer measuring the strength of the hand grip (measurement error 0.5 kg/1lb) were used to test the parameters of motor coordination and grip strength, and comprised a safe and comfortable glove (with sensitive electro-optical sensors for evaluating position, wrist speed, and finger movement; power supply: voltage: 5 V DC; rated current input: 300 mA), and the Medi Tutor TM software. The Hand Tutor was used to measure the kinematic parameters as follows: maximum range of movement (ROM) from flexion to extension (sensitivity: 0.05 mm of wrist and fingers Ext./Flex) and the frequency of movement (motion capture speed: up to 1 m/s) [37]. The system (MediTouch, Israel) is used by many leading physical and occupational therapy centers globally and has CE and FDA certification [38]. The Hand Tutor glove was worn on the hand of the directly affected side in stroke patients.

The study of the affected dominant and non-dominant hand and wrist motor coordination and grip strength was performed in two different starting positions: sitting (without support) and lying with stabilization of the examined upper limb. During the first examination, the subject sat on a treatment table, with feet resting on the floor. The upper limb was examined in adduction, with a bent (90 degrees) elbow joint in the intermediate position between pronation and supination of the forearm. In the supine examination, the upper limb was stabilized against the subject’s body (adduction in the shoulder joint, flexion in the elbow joint, 90 degrees, in an intermediate position). In each of the starting positions, they were asked to make movements as quickly as possible and to the full extent. Maximum range of motion (max ROM) was automatically calculated based on the full range of active motion from flexion to extension. The hand-grip strength was measured with a dynamometer in both analyzed starting positions after examining the frequency and maximal ranges of motion. In patients after a stroke, the right or left limb with paresis was examined. Establishment of the dominant limb was based on an interview.

Before starting the study, the patients were informed about the purpose of the study. They gave their consent in writing. Before each new task, the patient was informed about how to perform the task.

### 2.5. Sample Size Calculation

The sample size was estimated using the G* Power 3.1.9.4 program, assuming the following parameters: effect size d = 0.78 (parameter calculated on the basis of group averages for wrist mobility in the non-stabilized position); α = 0.05; power = 0.95 for the Wilcoxon–Mann–Whitney test; and the required sample size was 76 (38 people per group).

### 2.6. Statistical Analysis

All statistical analyzes were performed using IBM SPSS Statistics 25.0 (IBM Corporation, Armonk, NY, USA). In order to compare the two groups in terms of the analyzed parameters, analyzes were carried out using the Mann–Whitney U test. To compare two measurements, the Wilcoxon test was used. The level of significance was α = 0.05.

## 3. Results

The Mann–Whitney U test was used to compare patients with the dominant or non-dominant hand under test, in a sitting position without support and in a lying position with a stabilized upper limb.

In the sitting position, patients for whom the non-dominant hand was examined had higher scores only for max ROM F3 and F4 compared to patients with the dominant hand (Table 3). In contrast, lying with stabilization of the affected upper limb resulted in an improvement in most of the parameters studied: frequency of wrist and finger movements from 2 to 5, and max ROM of fingers from 5 to 3, in addition to hand grip strength (Table 4).

Using the Wilcoxon test, the results of parameters measured in the sitting and lying positions were compared among patients with the dominant hand or the non-dominant hand.

The analysis showed that the patients with the dominant hand had higher scores for the max ROM of the wrist in the lying position, and lower scores for max ROM F4 in the sitting position (Table 5). In contrast, the examination of the non-dominant hand showed that, in the sitting position without support, less significant results were achieved and scores were lower than those in the lying position. In addition, significantly more significant results with a greater value were achieved compared to those in the dominant hand in the lying position (Table 6).

## 4. Discussion

The results of the study showed that the improvement in coordination and grip strength in patients after stroke was clearly better when the trunk was in a lying position and the affected non-dominant upper limb was stabilized.

Motor coordination was assessed using the Hand Tutor TM, as in Carmela et al. and the earlier works of the authors of this study [38,39,40]. For the functional status analysis, commonly accepted scales and tests were used, such as MAS, Trunk Control Test, and the Fugl-Meyer UE assessment [34,35,41].

The aim of the research by Xiang et al. was to assess the influence of the hand on the accuracy of movement, and to compare 3D kinematic data relating to the performance of the dominant and non-dominant hand due to the influence of movement speed and target location. Dominant hand movements and/or brisk movements were more efficient with a straighter hand path and less torso rotation, but the dominant hand movements were less effective with rapid movements [42].

Armstrong et al. compared the strength of the dominant and non-dominant hand in right-handed and left-handed people. In all tests, no significant differences were observed between the hands of left-handed people, whereas small but significant differences were observed between the hands of right-handed people. In addition, there was considerable variability in the relative strength of both hands for each participant [31].

In 2019, El-Gohary et al., assessed different baseline positions in clamping force and keystroke testing of healthy participants (90° elbow flexion, 90° arm flexion, and dangling arm position). However, the comparison of the grip strength of the right and left hands did not show any significant differences. In contrast, the strength used to strike the key showed a significant increase in favor of the dominant hand [32].

In our study, the influence of lying position with stabilization of the upper limb on the parameters of motor coordination and grip strength in both dominant and non-dominant limbs was investigated. In this study, the paresis, non-dominant upper limb, in both sitting and lying positions, showed more significant results compared to the affected dominant upper limb. The comparison of the effects of the sitting and lying position on the dominant and non-dominant hand using the Mann–Whitney U test showed a significant advantage of the non-dominant hand in the lying position. In this case, significantly higher results were observed for the frequency of movements of both the wrist and fingers from F2 to F5, significantly greater maximum ranges of finger movement from F3 to F5, and significantly higher grip strength of the non-dominant hand. Similarly, the research of Ardon et al. assessed children with congenital hand differences using a functional grip test and body function measurements that included joint mobility and muscle strength. The authors found a stronger relationship between body functions and the ability to perform manual activities in non-dominant hands [43]. The same authors concluded that the improvement in body function leads to larger changes in the manual dexterity of the non-dominant hand. In turn, in the studies of Okunribidi et al., stabilization of the forearm was especially important for achieving greater grip strength, but these studies were carried out on healthy people [44]. Planning to adopt a movement posture, and movement planning and the preparation and execution of a target-oriented movement, are difficult for a person with a stroke. Yang Cl et al. showed that, after a stroke, when reaching with a paresis hand, trunk compensation occurs, which is characterized by greater rotation of the trunk and pelvis compared to reaching with the arm without paresis and in healthy control subjects [33].

In our study, the comparison of the results of the dominant and non-dominant hand in the examined positions using the Wilcoxson test showed more significant results for the non-dominant hand, especially in terms of frequency of movement (Hz 2 to 5 fingers) and wrist max ROM, whereas the dominant hand showed one result that was significantly higher in a lying position and one that was significantly higher in a sitting position. Our research proved that a lying position with stabilization of the upper limb is particularly beneficial for improving motor coordination and the grip strength of the non-dominant hand. We speculated that this may be because of an innate weakness in the muscles of the non-dominant limb. For example, Hoshiyama et al. showed that a specific interaction between signals after electrical stimulation and the activities of the sensory cortex related to writing with a non-dominant hand occurred in both hemispheres, whereas it was only recognized in the opposite hemisphere to the writing hand when using the dominant hand. It seems, therefore, that the somatosensory cortex was more activated, and thus it interacted with the applied stimulation during the unskillful movement of the non-dominant hand compared to the movement of the dominant hand [5]. Clinicians should strive to maintain and restore nearly equal dominant and non-dominant hand grip strength scores to ensure better hand function [45,46]. It also seems important to look for exercises and positions for exercises that will improve the parameters of, among other factors, movement coordination or the strength of the grip.

Overall, the results of our work show that the supine position with stabilization of the affected upper limb is particularly beneficial for improving motor coordination in the “weaker” (non-dominant) limbs.

Research value

Our study showed that the supine position with stabilization of the affected upper limb helps to improve coordination and grip strength of the distal, non-dominant upper limb, and may improve the function and activity of stroke patients.

Study limitation

A limitation of the study was the analysis only of patients after ischemic stroke and in the acute phase of the disease. In addition, the number of participants in the study was lower than our simple size calculations indicated, but we used the formula that was supposed to provide the most computing power. The difference in the number of people tested was eight in each group. Another limitation was the examination of only the hand with dominant or non-dominant paresis. However, this study also emphasized that the supine position with stabilization of the affected upper limb helps to achieve better coordination and strength parameters of the weaker parts of our body.

Clinical messages

Stabilization of the shoulder against the trunk in the supine position may improve the coordination and grip strength, especially of the non-dominant distal part of the upper limb in patients after ischemic stroke.

Clinicians should strive to maintain and restore comparable coordination and grip strength scores for both the dominant and non-dominant hands to ensure better hand function in daily bilateral activities.

## 5. Conclusions

Hand and wrist coordination results and hand-grip strength are significantly higher in the supine position of the trunk and stabilization of the affected, non-dominant upper limb.

A lying position with stabilization of the affected upper limb has a particularly beneficial effect on the coordination of movements and the grip strength of the weaker, non-dominant hand.

## Figures and Tables

**Figure 1 brainsci-12-00164-f001:**
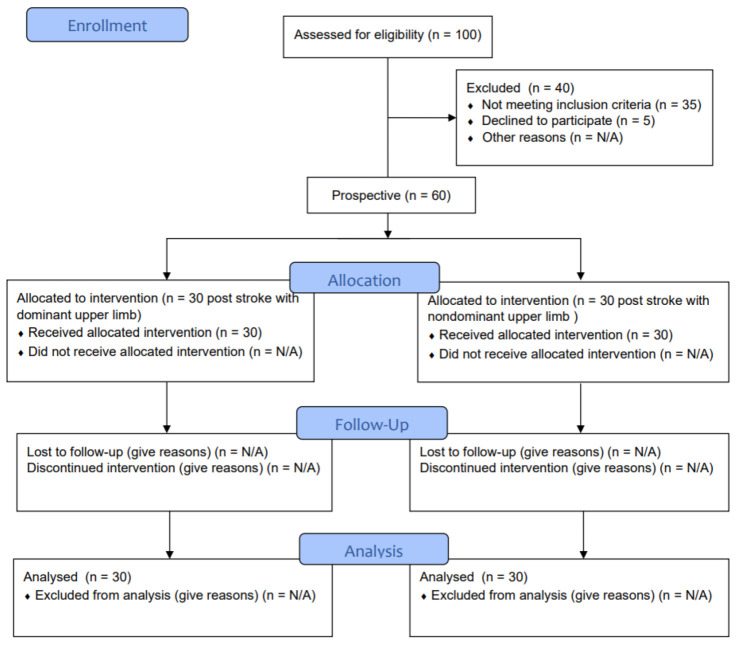
Consort flow diagram.

**Table 1 brainsci-12-00164-t001:** The basic epidemiological data of the study population and the clinical control group.

Total Number of Participants N = 60 (100%)	Post-Stroke Group *n* = 60 (100%)	
Sex	Female	Male
n/%	30 (50%)	30 (50%)
Cerebral ischemic stroke (thromboembolic) n/%	60 (100%)	
Time post stroke/episode (week)	5–8	
Right affected side	30 (50%)	
Left affected side	30 (50%)	
Dominant right hand	60 (100%)	
Dominant left hand	N/A	

**Table 2 brainsci-12-00164-t002:** Biometric data of the study population and the clinical control group.

Group	Age	Height	Body Mass	BMI
Dominant hand	64.87 ± 16.29	170.07 ± 9.12	73.97 ± 8.18	25.55 ± 1.76
Non-dominant hand	65.80 ± 12.86	171.73 ± 7.76	78.07 ± 10.81	26.41 ± 2.83
Mann Whitney U	445.0	377.0	357.0	383.5
Z	−0.07	−1.38	−1.08	−0.98
*p*	0.941	0.169	0.279	0.325
effect size	0.01	0.18	0.14	0.13

**Table 3 brainsci-12-00164-t003:** Comparison of parameters in the sitting (unstable) position in patients with stroke in whom the dominant or non-dominant hand was examined.

	DOMINANT			NON-DOMINANT					
	M	Me	SD	M	Me	SD	Z	*p*	r
Hz wrist. [cycle/sec]	0.99	0.80	0.68	1.30	1.10	0.74	−1.93	0.054	0.35
Wrist MaxROM [mm]	15.91	14.50	9.37	16.59	17.70	5.62	−1.06	0.287	0.19
Hz F5	1.26	1.05	0.74	1.76	1.80	0.98	−1.90	0.057	0.25
F5 MaxROM	15.88	16.60	7.72	19.63	19.60	8.39	−1.69	0.092	0.22
HzF4	1.27	1.05	0.74	1.73	1.70	1.01	−1.68	0.093	0.22
F4 MaxROM	20.19	20.20	6.72	23.66	23.75	8.41	−1.96	0.050	0.25
HzF33	1.27	1.05	0.74	1.72	1.70	1.00	−1.64	0.100	0.21
F3 MaxROM	19.87	19.40	5.37	22.38	23.50	5.36	−2.07	0.038	0.27
HzF2	1.27	1.05	0.74	1.72	1.70	0.99	−1.70	0.088	0.22
F2 MaxROM	16.70	17.25	5.33	18.86	18.75	5.09	−1.72	0.085	0.31
HzF1	1.09	0.90	0.82	1.40	1.10	1.00	−1.13	0.257	0.15
F1 MaxROM	7.75	6.95	5.65	10.62	9.65	5.54	−1.89	0.059	0.24
Grip strength [kg]	14.73	14.75	8.99	22.97	21.90	15.31	−1.94	0.053	0.25

Legend. M—mean; Me—median; SD—standard deviation; the result of the Z test, *p* significance level; r the correlation coefficient.

**Table 4 brainsci-12-00164-t004:** Comparison of parameters in a lying (stabilized) position in patients with stroke in whom the dominant or non-dominant hand was examined.

	DOMINANT			NON-DOMINANT					
	M	Me	SD	M	Me	SD	Z	*p*	r
Hz Wrist. [cycle/sec]	0.93	0.70	0.67	1.31	1.30	0.85	−2.13	0.033	0.39
MaxROM [mm]	19.69	20.00	9.82	20.62	22.00	5.47	−1.41	0.160	0.26
HzF5	1.36	1.15	0.79	1.93	1.85	1.02	−2.34	0.019	0.43
F5 MaxROM	14.67	13.75	7.71	19.73	19.10	9.99	−2.13	0.033	0.39
HzF4	1.30	1.05	0.82	1.93	1.85	1.03	−2.55	0.011	0.46
F4 MaxROM	18.25	17.65	6.71	21.50	21.90	6.24	−2.08	0.038	0.38
HzF3	1.37	1.15	0.80	1.93	1.85	1.03	−2.26	0.024	0.41
F3 MaxROM	18.49	17.65	4.83	21.17	20.70	5.02	−2.09	0.036	0.38
HzF2	1.37	1.15	0.81	1.93	1.85	1.03	−2.26	0.024	0.41
F2 MaxROM	15.95	15.85	5.17	18.40	18.15	4.49	−1.80	0.071	0.33
HzF1	1.15	0.90	0.92	1.25	1.05	0.95	−0.49	0.625	0.09
F1 MaxROM	7.23	6.75	5.23	9.13	8.70	4.48	−1.53	0.126	0.28
Grip strength [kg]	15.29	13.20	9.90	23.62	22.15	14.67	−2.39	0.017	0.44

Legend. M—mean; Me—median; SD—standard deviation; the result of the Z test, *p* significance level; r the correlation coefficient.

**Table 5 brainsci-12-00164-t005:** Comparison of the results of the parameters in the sitting (unstable) and lying (stabilized) positions among patients in which the dominant hand was examined.

	NON-STABILIZED			STABILIZED					
	M	Me	SD	M	Me	SD	Z	*p*	r
Hz Wrist. [cycle/sec]	0.99	0.80	0.68	0.93	0.70	0.67	−0.43	0.665	0.06
Wrist MaxROM [mm]	15.91	14.50	9.37	19.69	20.00	9.82	−3.62	<0.001	0.47
HzF5	1.26	1.05	0.74	1.36	1.15	0.79	−1.11	0.267	0.14
F5 MaxROM	15.88	16.60	7.72	14.67	13.75	7.71	−1.16	0.245	0.15
HzF4	1.27	1.05	0.74	1.30	1.05	0.82	−0.66	0.507	0.09
F4 MaxROM	20.19	20.20	6.72	18.25	17.65	6.71	−2.66	0.008	0.34
HzF3	1.27	1.05	0.74	1.37	1.15	0.80	−1.18	0.237	0.15
F3 MaxROM	19.87	19.40	5.37	18.49	17.65	4.83	−1.90	0.057	0.25
HzF2	1.27	1.05	0.74	1.37	1.15	0.81	−1.18	0.237	0.15
F2 MaxROM	16.70	17.25	5.33	15.95	15.85	5.17	−1.26	0.209	0.16
HzF1	1.09	0.90	0.82	1.15	0.90	0.92	−0.85	0.396	0.11
F1 MaxROM	7.75	6.95	5.65	7.23	6.75	5.23	−0.50	0.614	0.07
Grip strength [kg]	14.73	14.75	8.99	15.29	13.20	9.90	−0.93	0.352	0.12

Legend. M—mean; Me—median; SD—standard deviation; the result of the Z test, *p* significance level; r the correlation coefficient.

**Table 6 brainsci-12-00164-t006:** Comparison of the results of the parameters in the sitting (unstable) and lying (stabilized) positions among patients in which the examined hand was the non-dominant hand.

	NON-STABILIZED			STABILIZED					
	M	Me	SD	M	Me	SD	Z	*p*	r
Hz Wrist. [cycle/sec]	1.30	1.10	0.74	1.31	1.30	0.85	−0.05	0.959	0.01
Wrist MaxROM [mm]	16.59	17.70	5.62	20.62	22.00	5.47	−3.46	0.001	0.45
HzF5	1.76	1.80	0.98	1.93	1.85	1.02	−2.48	0.013	0.32
F5 MaxROM	19.63	19.60	8.39	19.73	19.10	9.99	−0.68	0.497	0.09
HzF4	1.73	1.70	1.01	1.93	1.85	1.03	−2.55	0.011	0.33
F4 MaxROM	23.66	23.75	8.41	21.50	21.90	6.24	−2.21	0.027	0.29
HzF3	1.72	1.70	1.00	1.93	1.85	1.03	−2.85	0.004	0.37
F3 MaxROM	22.38	23.50	5.36	21.17	20.70	5.02	−2.13	0.033	0.27
HzF2	1.72	1.70	0.99	1.93	1.85	1.03	−2.79	0.005	0.36
F2 MaxROM	18.86	18.75	5.09	18.40	18.15	4.49	−0.95	0.341	0.12
HzF1	1.40	1.10	1.00	1.25	1.05	0.95	−0.16	0.876	0.02
F1 MaxROM	10.62	9.65	5.54	9.13	8.70	4.48	−2.36	0.018	0.30
Grip strength [kg]	22.97	21.90	15.31	23.62	22.15	14.67	−1.09	0.276	0.14

Legend. M—mean; Me—median; SD—standard deviation; the result of the Z test, *p* significance level; r the correlation coefficient.

## Data Availability

Data available on request from corresponding author.

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
