# Peer review of "Motor Coordination and Grip Strength of the Dominant and Non-Dominant Affected Upper Limb Depending on the Body Position—An Observational Study of Patients after Ischemic Stroke"

_brainsci, 2022, doi:10.3390/brainsci12020164_

Round 1

Reviewer 1 Report
Thank you for the opportunity to review this interesting article. Please, respond my concerns:
ABSTRACT:
-The firsts two sentences must be a short background
-In methods, line 15, please use "acute ischemic stroke"
-"Main measures:". Please add fluency to the sentence.
INTRODUCTION:
-It would be necessary define ischemic stroke.
-The introduction is correct and informative.
METHODS:
-What kind of observational study?
-Please, figure 1 in one page.
RESULTS:
-Please, change commas by points.
-Tables are very clear and results well written.
DISCUSSION
-Discussion "study value", "limitations" and "clinical messages" should appear without include subsections. 
-Please, do not use point for each clinical message. This style is not MDPI.C
CONCLUSIONS:
-Please, add more information.
REFERECES:
-Adequated.

Author Response

Manuscript ID: brainsci-1555408
Type of manuscript: Article
Title (previous): Motor coordination and grip strength of the dominant and non-dominant affected upper limb depends on the body position – an observational study of post-ischemic stroke patients.
Title (after change): Motor coordination and grip strength of the dominant and non-dominant affected upper limb depending on the body position - an observational study of patients after ischemic stroke.

Dear Reviewers,
       Thank you very much for the analysis of our manuscript. We really appreciate your comments and indication of fragments that should be corrected and explained. Considering your suggestions, all mistakes were corrected. The introduction of corrections and changes in the text caused the numbering of the lines to shift. In response to reviewers' comments, it provides the original numbering. In order to avoid misunderstandings, changes introduced in the text are marked in blue and additionally, the manuscript was sent in the change tracking mode.

Reviewer #1:

Thank you for showing interest in this manuscript and I am delighted to find it interesting.
Thank you very much for that.
Regarding the remarks and comments, I am writing back.
The following comments and answers:
ABSTRACT:
The firsts two sentences must be a short background
In methods, line 15, please use "acute ischemic stroke"
"Main measures:". Please add fluency to the sentence.

The summary has been redrafted in line with your suggestions. I introduced a short background for the presented topic of work. I edited the sentence concerning the characteristics of patients after ischemic stroke by adding "acute ischemic stroke". Due to the addition of a short background, the suggested lines are inconsistent ("In methods, line 15, please use" acute ischemic stroke "), now the corrected sentence is in line 18. Moreover, I improved the fluency of the main measures.
Thank you very much for all suggestions for a summary of the work.

INTRODUCTION:
It would be necessary define ischemic stroke.
The introduction is correct and informative.

As suggested, in the introduction I have characterized the ischemic stroke (lines 84 to 87).
Thank you for this tip and for your appreciation for the introduction section.

METHODS:
What kind of observational study?
Please, figure 1 in one page.
The study presented is an observational, prospective study.
I have provided an appropriate supplement in the summary of the work and the study design.
Thank you for this suggestion.
Figure 1, as suggested, I have moved to one side (page 4).
Thank you for pointing out these shortcomings.

RESULTS:
Please, change commas by points.
Tables are very clear and results well written.

In the Results section, I changed the commas to points.
Thank you for your comment and for your appreciation of this section.

DISCUSSION
Discussion "study value", "limitations" and "clinical messages" should appear without include subsections.
Please, do not use point for each clinical message. This style is not MDPI.C

Indeed, I scored inadequately for "study value", "limitations" and "clinical messages". In the current form, there are no sub-items in the discussion section.
Thank you for this comment.

CONCLUSIONS:
Please, add more information.

Your request for more information, in conclusion, seems quite difficult to me. However, I tried to highlight what was most important in the results of this study and decided to add the following sentence: "Hand and wrist coordination results and handgrip strength are significantly higher in the supine position of the trunk and stabilization of the affected, non -dominant upper limb ".
Thank you very much for pointing this out.

REFERECES:
Adequated.

Thank you very much for your appreciation.

In addition, throughout the text, taking into account the suggestions of the second reviewer, there are changes to the determination of the items to be tested and a slight correction of the title of the work, which makes the message more consistent.
Thank you very much for the thorough analysis of our manuscript.
Thank you very much for your time.

Reviewer 2 Report
There are two main difficulties in this paper about the influence of trunk stabilization on hand grip.
First, the authors do not take into account the literature about the relation of hand and trunk movement that have been previously published in stroke patients. This is the basis of the knowledge on this topic and has to be introduced at least in the introduction and in the discussion related to their results. There work have to be positioned among this literature that  have explored this topic. The very large view of the introduction can be summarized and completed by existing results in stoke patients.
Second the authors confuse the recumbent position with trunk stabilization. Authors have compared upper limb movement in the recumbent and lying positions. But these two positions are not comparable due to the strong differences of visual and perceptive inputs. The difficulty is that it seems to be an error of methodology due to the lack of knowledge of previous published papers on postural control of the trunk  in stroke patients. The authors must rewrite their paper in a way to compare hand grip in these two positions and NOT comparing stabilized and unstabilized movement neither in the introduction nor in the discussion.
It is not enough to write in the title that hand grip depends on the body position, it has to be clearly justified in the introduction to lead to this very specific paradigme. It has also to be discussed in the discussion.

Author Response

Manuscript ID: brainsci-1555408
Type of manuscript: Article
Title (previous): Motor coordination and grip strength of the dominant and non-dominant affected upper limb depends on the body position – an observational study of post-ischemic stroke patients.
Title (after change): Motor coordination and grip strength of the dominant and non-dominant affected upper limb depending on the body position - an observational study of patients after ischemic stroke.

Dear Reviewers,
Thank you very much for the analysis of our manuscript. We really appreciate your comments and indication of fragments that should be corrected and explained. Considering your suggestions, all mistakes were corrected. The introduction of corrections and changes in the text caused the numbering of the lines to shift. In response to reviewers' comments, it provides the original numbering. In order to avoid misunderstandings, changes introduced in the text are marked in blue and additionally, the manuscript was sent in the change tracking mode.

Reviewer #2:
Thank you very much for the very quick and thorough analysis of our manuscript.
Regarding the remarks and comments, I am writing back.
The following comments and answers:

There are two main difficulties in this paper about the influence of trunk stabilization on hand grip.
First, the authors do not take into account the literature about the relation of hand and trunk movement that have been previously published in stroke patients. This is the basis of the knowledge on this topic and has to be introduced at least in the introduction and in the discussion related to their results. There work have to be positioned among this literature that  have explored this topic. The very large view of the introduction can be summarized and completed by existing results in stoke patients.

In response to your above comment, I have appropriately supplemented the introduction and discussion. As a result, eight new cited items appeared in the bibliography and I added them to my bibliography, which increased the reference to 49 items. Added items:

  1. Liao, C.F.; Liaw, L.J.; Wang, R.Y.; Su, F.C.; Hsu, A.T. Relationship between trunk stability during voluntary limb and trunk movements and clinical measurements of patients with chronic stroke. J. Phys. Ther. Sci. 2015, 27, 2201–2206, doi:10.1589/jpts.27.2201.
  2. Brunnstrom, S. Movement Therapy in Hemiplegia: A Neurophysiological Approach; Harper and Row: New York, NY, USA, 1970.
  3. Ellis, M.D.; Sukal-Moulton, T.; Dewald, J.P. Progressive shoulder abduction loading is a crucial element of arm rehabilitation in chronic stroke. Neurorehabilit. Neural Repair 2009, 23, 862–869.
  4. Adams HP, Bendixen BH, Kappelle LJ, Biller J, Love BB, Gordon DL, Marsh EE. Classification of subtype of acute ischemic stroke. Definitions for use in a multicenter clinical trial. TOAST. Trial of Org 10172 in Acute Stroke Treatment. Stroke. 1993 Jan;24(1):35-41.
  5. Armstrong, C.A.; Oldham, J.A. A Comparison of Dominant and Non-Dominant Hand Strengths. J. Hand Surg. 1999, 24, 421–425.
  6. El-Gohary, T.M.; Abd Elkader, S.M.; Al-Shenqiti, A.M.; Ibrahim, M.I. Assessment of hand-grip and key-pinch strength at three arm positions among healthy college students: Dominant versus non-dominant hand. J. Taibah Univ. Med. Sci. 2019, 14, 566–571.
  7. Yang, C.L.; Creath, R.A.; Magder, L.; Rogers, M.W.; McCombe Waller, S. Impaired posture, movement preparation, and execution during both paretic and nonparetic reaching following stroke. J. Neurophysiol. 2019, 121, 1465–1477.
  8. Yang, C.L.; Creath, R.A.; Magder, L.; Rogers, M.W.; McCombe, Waller S. Impaired posture, movement preparation, and execution during both paretic and nonparetic reaching following stroke. J Neurophysiol. 2019, 121(4):1465-77.

Due to the addition of new references, it was necessary to enter a new citation numbering in line with the newly added references in the reference list at the end of the manuscript.
Thank you very much for this important suggestion.

Second the authors confuse the recumbent position with trunk stabilization. Authors have compared upper limb movement in the recumbent and lying positions. But these two positions are not comparable due to the strong differences of visual and perceptive inputs. The difficulty is that it seems to be an error of methodology due to the lack of knowledge of previous published papers on postural control of the trunk  in stroke patients. The authors must rewrite their paper in a way to compare hand grip in these two positions and NOT comparing stabilized and unstabilized movement neither in the introduction nor in the discussion.

Indeed, I misplaced the text, emphasizing the differences in testing with regard to trunk stabilization and its absence. In fact, what I meant was in which position the body is more or less stable when lying down/sitting. And yet I would like to draw attention to the differences in the results of motor coordination and the pressure force of the diseased dominant and non-dominant upper limb in the various positions examined, sitting and lying with the upper limb next to the patient's body. Of course, I corrected the text, highlighting the positions in which patients were examined after a stroke. This correction resulted in a change in the title of the work, a correction of the abstract, as well as individual sections of the work.
Thank you very much for this valuable suggestion.

It is not enough to write in the title that hand grip depends on the body position, it has to be clearly justified in the introduction to lead to this very specific paradigme. It has also to be discussed in the discussion.

As I wrote earlier, I made a slight correction of the title of the work, which does not change the fact that the analysis concerned the assessment of the results of movement coordination and the strength of the grip of the affected dominant and non-dominant hand in patients after ischemic stroke. Therefore, in the introduction to the work and in the discussion I introduced new citations, which I hope complement the knowledge in this topic. The suggested redrafting of the work resulted in the appearance of changes throughout the work. Moreover, the citations (lines 288 to 292, 308 to 315, and 330 to 334), in the discussion seem to better represent the topic.

Thank you very much for your thorough analysis of our work.

Undoubtedly, the subject requires further in-depth research.
Thank you for all the critical comments and the opportunity to broaden my knowledge.
Thank you very much for your all-important comments and suggestions.
Thank you very much for your time.